# Optimization of 3D printing supply chain in the era of live streaming e-commerce

**Zhen Chen**[1], **Ying Tang**[2]*

**1** Guangdong Coastal Economic Belt Development Research Center, Lingnan Normal University, Guangdong, China, **2** Business School, Lingnan Normal University, Guangdong, China

\* tangying9590@163.com

## Abstract

This study examines the effects of the rising live streaming e-commerce on the 3DP supply chain, employing system dynamics to develop separate models for pure polymer and polymer-metal mixed printing. The analysis focuses on optimizing the 3DP supply chain configuration. Results indicate that, based solely on printing time, cost, and quality metrics, Corporate-live-3DP services are optimal for live commerce scenarios. However, despite this, Private-live-3DP maintains a substantial consumer base in practice, as evidenced by literature data and case studies. Both models pose significant challenges to conventional supply chains, necessitating adaptation. For Corporate-live-3DP, optimization strategies may include technology advancements, digital transformation, agile manufacturing, global network optimization, innovative management, collaborative R&D, fine-tuned inventory control, quality system upgrades, talent development, and organizational restructuring. Conversely, Private-live-3DP can be optimized through consolidation of private 3D printing resources, demand prediction and order optimization, supply chain collaboration platforms, quality management extensions, inventory strategy adjustments, increased transparency, regulatory compliance, and risk mitigation measures.

## 1. Introduction

### 1.1 3DP development history

3D printing (3DP) technology creates physical objects directly from digital design files by stacking materials layer by layer [1]. Compared with traditional subtractive manufacturing methods, 3DP generates less material consumption and its waste can also be recycled. It embodies its environmental friendliness and is a circular technology. Since 2016, the appetite for employing 3DP in the production of consumer goods has burgeoned notably. Testament to this growth, the worldwide 3DP market reached a valuation of 18 billion USD in 2022. In 2016, this figure was only 6 billion, 200% growth in 8 years [2]. 3DP has significant potential to enhance the resilience, flexibility, and localization of the global economic system [3]. The details are as follows.

- Elastic

**Data Availability Statement:** All relevant data are within the manuscript.

**Funding:** 1. Guangdong Philosophy and Social Science Planning Project, grant number GD22YDXZYJ02. 2. Special projects in key fields of Guangdong Provincial Department of Education,

grant number 2021ZDZX3003. The funders had no role in study design, data collection and analysis, decision to publish, or preparation of the manuscript.

**Competing interests:** The authors have declared that no competing interests exist.

Rapid response and emergency manufacturing: In the face of natural disasters, supply chain disruptions, or other emergencies, traditional manufacturing relies on complex supply chain networks and large-scale production facilities, which may become fragile during crises. In contrast, 3DP can quickly respond to changes in demand, redesign and print out required items in a short period of time, such as medical equipment, temporary housing components, or critical components. For example, during the COVID-19 pandemic, 3DP was widely used for the rapid manufacturing of personal protective equipment (PPE), ventilator components, and virus detection equipment to cope with the global shortage of medical supplies [4–7].

Distributed production capacity: 3DP technology enables the production process to be dispersed to various corners, no longer limited to centralized factories. This distributed production capability helps to avoid supply disruptions caused by a single point of failure. When an emergency occurs in a certain region, 3DP facilities in other areas can quickly fill the supply gap, thereby enhancing the overall system's resilience. For example, medical institutions or community centers in remote areas are equipped with 3D printers, which can print medical equipment locally and reduce reliance on long-distance transportation [8–11].

- Flexibility

On demand production and personalized customization: Traditional mass production often requires large-scale orders to dilute fixed costs, leading to inventory backlog and waste. 3DP can adjust production according to actual needs at any time, achieving single piece or small batch customization without the need for additional molds or high minimum order quantities. For example, the fashion industry uses 3DP to create customized shoes, allowing consumers to choose their own designs and sizes. Manufacturers only start printing after receiving orders, reducing inventory risks and resource waste [12–14].

Rapid prototyping and iterative design: In the product development process, 3DP greatly shortens the time from design to physical verification, enabling enterprises to quickly iterate product design, test performance, and timely push it to the market. For example, engineers in the aerospace and automotive industries can quickly print prototypes of complex components for functional testing without waiting for lengthy mold manufacturing and traditional manufacturing processes, accelerating innovation processes and maintaining competitive advantages [15,16].

- Localization

Reducing logistics dependence and carbon footprint: 3DP can be directly manufactured where items are needed, reducing the demand for long-distance transportation, lowering logistics costs and carbon emissions. This is particularly beneficial for remote areas, developing countries, or environments with limited resources. For example, some villages in Africa use 3DP technology to manufacture agricultural tools, water purification equipment and other daily necessities, reducing their dependence on imported goods and promoting local economic self-sufficiency [17,18].

Enabling grassroots innovation and entrepreneurship: 3DP has lowered the innovation threshold, enabling small enterprises and individual entrepreneurs to use local resources to manufacture products and promote local innovation and economic development. For example, community maker spaces or small studios can use 3D printers to manufacture art installations, home furnishings, or electronic products that not only meet local market demand, but may also go global through e-commerce platforms, promoting local economic diversification [19–21].

## 1.2 Live streaming e-commerce and 3DP

Live streaming e-commerce, an emerging retail format, began to gain global prominence around the mid-2010s, experiencing exponential growth particularly in the Chinese market before gradually influencing and spreading to other regions. Through empirical surveys of consumers, factors such as anchor characteristics, interaction mechanisms, and trust building in live streaming e-commerce were established to affect consumer purchase intentions, and for understanding the success factors of live streaming e-commerce. Empirical analysis has revealed the relationship between audience engagement and purchase intention in live streaming e-commerce, emphasizing the important role of real-time interaction, social atmosphere, and emotional resonance in enhancing the attractiveness of live streaming shopping [22,23].

The confluence of digital technologies and e-commerce has broadened the horizons of 3DP applications and fundamentally transformed supply chain dynamics [24]. Notably, the pandemic era has witnessed the ascent of live streaming commerce, further expanding the scope for 3DP integration. As individuals confined to their homes yearn for swift access to gifts, digital peripherals, unconventional emergency spares, or bespoke items, the agility and precision offered by 3DP customization models have proven ideal in fulfilling these diverse consumer demands [25–27]. This agile production method enables the swift and precise fabrication of objects in myriad forms and tailored to individual preferences, ensuring that consumers can acquire their coveted items within the shortest feasible timeframe.

Taking China as an example, Douyin (http://www.douyin.com) is the Chinese version of the high-profile social platform Tiktok, with its core content focused on live streaming social interaction. Ocean engine is a live big data index query website (https://trendinsight. oceanengine.com) supported by Douyin. Through this website, we can gain insight into the search trends with the keywords of "3D printing" and "customization" in the past five years in China. Data shows that the demand for customized products among consumers is on the rise, and this trend is becoming increasingly significant. Since 2021, live broadcasts involving 3DP have begun to appear on the platform, and the trend is also increasing year by year. Table 1 details the search trends of the above keywords over the past five years, further confirming the huge potential of this market.

## 1.3 Impact of 3DP on the supply chain

The rapid ascent of Live streaming e-commerce is catalyzing a transformative shift in the 3DP supply chain. Consumers, deeply engaged in real-time shopping, increasingly seek 3DP goods differing by design, function, and composition. Specifically, the presence or absence of metallic elements. This dichotomy spawns diverse demands for printers, materials, and infrastructure to efficiently fulfill tailored orders.

To achieve 3DP, desktop printers and industrial printers can be used. But their applicable materials are different, which will affect the design and optimization of the supply chain process. There are two mainstream materials, metal and polymer. The same 3D printer cannot

**Table 1. Search popularity of 3DP and customization on Mainstream Live E-commerce in China.**

| Year | Search Index of 3DP | Search Index of Customization |
|------|--------------------|-------------------------------|
| 2019 | - | 42000 |
| 2020 | - | 63000 |
| 2021 | 2445 | 26000 |
| 2022 | 5785 | 59000 |
| 2023 | 34000 | 539000 |

print metal and polymer materials simultaneously because the principles of metal and polymer 3DP processes are different. The former mostly adopts selective laser sintering (SLS) or electron beam melting (EBM) technologies, using a metal powder for its printing material, while the latter adopts a stereolithography apparatus (SLA) or fused deposition modeling (FDM) technologies, with a polymer filament printing material.

Industrial-grade printers, with their proficiency in printing metals and polymers at elevated standards, cater aptly to the requirements of corporate live streamers and remote customization endeavors [28]. By industry definition, a device priced above $5000 falls under the industrial printer category. Such equipment and associated printing solutions are characterized by advanced maturity, having secured the necessary approvals from relevant stakeholders without entanglements in intellectual property disputes—albeit at a substantially higher cost [29,30].

Conversely, desktop printers predominantly process polymer materials, offering affordability and compatibility with the needs of DIY enthusiasts, independent live streamers, and localized manufacturing setups. Classified as such when priced below $5000, these printers largely rely on open-source data, presenting a relatively low barrier to entry. They are favored by home users who predominantly utilize them for producing everyday consumer goods [31]. While unable to handle metal materials and lagging behind industrial printers in terms of print quality and speed, desktop printers compensate by granting users a heightened degree of creative autonomy and the satisfaction of immediate results [32].

## 1.4 Summary

This study examines the effects of the rising live streaming e-commerce on the 3DP supply chain, employing system dynamics to develop separate models for pure polymer and polymer-metal mixed printing. The analysis focuses on optimizing the 3DP supply chain configuration.

This study undertakes a meticulous analysis of disparities across 3DP offerings, probing production nuances, material sourcing, logistics, and fulfillment unique to metallic and non-metallic categories. It illuminates challenges and opportunities within the Live streaming e-commerce context, providing a foundation for actionable strategies to optimize 3DP supply chains for agility, efficiency, and responsiveness to market fluctuations.

In summary, this study propels 3DP's full potential within live streaming e-commerce by unraveling its distinct supply chain complexities and devising innovative solutions. It charts a course for streamlined, adaptive, and customer-focused 3DP operations, reshaping digital-era shopping, production, and consumption.

## 2. Modeling

### 2.1 Problem statement

On live streaming e-commerce platforms, both individuals and businesses have the capability to offer customized products to consumers using 3DP technology. While individuals typically rely on desktop printers for production, enterprises generally utilize industrial-grade printers. The evaluation of these 3DP products is based on their overall utility to consumers, regardless of whether they are customized by individuals or enterprises. This prompts the question:

**What is the most ideal 3DP customization mode on these platforms? Should it be desktop printing offered by private live teams or industrial printing provided by corporate live teams?**

This study found effective answers to the above questions through system dynamics methods. System dynamics is an effective tool for optimizing production process models. Because it has the following characteristics:

- Consider system integrity and dynamism

System dynamics focuses on how the system operates as a whole, rather than isolated components. It focuses on analyzing the interrelationships and feedback mechanisms between various elements in the production process, as well as the dynamic evolution of these relationships over time. This global perspective helps to reveal the nonlinear and delay effects hidden in complex processes, ensuring that the model can capture fluctuations, oscillations, stable states, or sudden changes in actual production systems.

- Quantitative modeling and computer simulation

System dynamics models typically adopt quantitative forms, transforming key variables of the production process into mathematical expressions and conducting computer simulation through specialized software. This simulation technology can simulate system responses under different conditions, conduct "hypothesis analysis", and predict the impact of different decisions or external shocks on system performance without actually changing the production environment. This is of great value for evaluating the effectiveness of policies, strategies, or technological improvements, as well as for risk management.

- Adaptability and customizability

Different production environments and industry characteristics vary, and the construction process of system dynamics models allows researchers to adjust the model structure and parameters based on the actual situation of specific enterprises. This high degree of adaptability and customizability enables the model to accurately reflect the unique logic of specific production processes, ensuring the pertinence and accuracy of simulation results.

In summary, system dynamics, with its in-depth characterization of system complexity and dynamic behavior, as well as powerful quantitative simulation and decision support capabilities, has become an ideal tool for constructing a 3DP production process model in this study. This helps to gain a deeper understanding of the internal operational mechanism of the 3DP supply chain in live streaming e-commerce, identify bottlenecks and opportunities, optimize resource allocation, formulate effective strategies, and improve overall operational performance.

By comparing the comprehensive utility of 3DP processes based on cost, quality, and time in different scenarios, determine a better optimal hybrid layout of desktop printing and industrial printing, where the consumer can maintain higher quality 3DP products in a shorter time and at a lower cost This could be a possible manufacturing scenario in the live streaming e-commerce era.

## 2.2 Scenario setting

At present, the mainstream printing materials used in consumer goods manufacturing are metals and polymers. As desktop printers can difficultly achieve metal printing, in this paper, we assume that all metal parts are only produced by industrial printers, while the polymer parts can be manufactured by both printer types.

**In this research, two printing modes are investigated: private-live-3DP and corporate-live-3DP.** Scenario 1 contains only the polymer printing process, while Scenario 2 contains a hybrid process consisting of polymer printing and metal printing. According to the printing process, a relevant system dynamics model is established, and the comprehensive utility of the two models was calculated and compared according to the three major standards of cost, quality, and time, in order to determine the feasible layout of 3DP in different scenarios. The process of modeling refers to the relevant literature [33–39].

The settings of the above scenarios are:

(1) 3DP differs from the traditional material subtractive manufacturing, and its material loss rate is very low. The material loss rate of the 3DP process is assumed zero.

(2) Neither scenario accounts for differences in means of transport. Transportation cost and time are determined only by transportation volume, distance, and unit price.

(3) Pertaining to the cost of 3DP, the purchase cost of the printing equipment is depreciated to the average printing cost of each finished product. Therefore, the printing cost relates only to the nature of printing materials, the number of finished products, and the printer type, while the amount of printing equipment is not considered. The printer type refers to a desktop printer or an industrial printer.

## 2.3 Scenario 1

In Scenario 1, only polymers are used to produce final products.

**2.3.1 Printing process.**    Table 2 displays the settings of participants in the printing process.

Step 1: The starting point of Scenario 1 is A.

Step 2: A raw polymer is transported from A to C and processed into polymer printing filaments. A filament is a common form of polymer printing consumables.

Step 3: There are different material flow routes for the modes of private-live-3DP and corporate-live-3DP.

In the private-live-3DP mode, the private live team purchases polymer printing filaments directly from C, prints and post-processes with the desktop printer to get the final products, and delivers it to the consumers through local express. Because the distance between private workshops and end consumers is very close, we set them all in E.

In the corporate-live-3DP mode, the corporate live team purchases the polymer printing filaments directly from C, prints and post-processes the final products using the industrial printers in F, and delivers the final product to E.

According to the flow routes, Fig 1 displays the system dynamics model of Scenario 1.

**Table 2. Participants in Scenario 1.**

| Symbol | Meaning | Symbol | Meaning |
|--------|---------|--------|---------|
| A | Raw polymer supplier | E | Private-live-3DP workshop/ End consumer |
| C | Polymer printing filament producer | F | Corporate-live-3DP workshop |

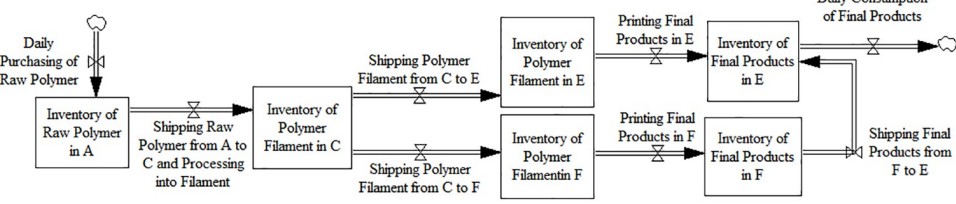

**Fig 1. System dynamics model of scenario 1.**

**2.3.2 Basic formulas.** According to Fig 1, the calculation formulas of cost, time, and quality of the routes are as follows.

- Cost

(1) Cost of private-live-3DP = Freight of a raw polymer from A to C + Processing fee of the raw polymer in C + Freight of the polymer filaments from C to E + Printing fee of the polymer parts in E + Post-processing and local express fee of the final products in E.

(2) Cost of corporate-live-3DP = Freight of a raw polymer from A to C + Processing fee of the raw polymer in C + Freight of the polymer filaments from C to F + Printing fee of the polymer parts in F + Post-processing fee of the final products in F + Freight of the final products from F to E.

- Time

(3) Time of private-live-3DP = Transport time of a raw polymer from A to C + Processing time of the raw polymer in C + Transport time of the polymer filaments from C to E + Printing time of the polymer parts in E + Post-processing and local express time of the final products in E.

(4) Time of corporate-live-3DP = Transport time of a raw polymer from A to C + Processing time of the raw polymer in C + Transport time of the polymer filaments from C to F + Printing time of the polymer parts in F + Post-processing time of the final products in F + Transport time of the final products from F to E.

- Quality

According to the above literature, the quality of the corporate-live-3DP product is at least as good and/or higher than the quality of the private-live-3DP product.

**2.3.3 Total utility calculation.** Since the supply chain process from A to C is the same for both printing modes in Scenario 1, we must compare the utilities of the private-live-3DP and corporate-live-3DP modes on the route of materials from C to E. Table 3 displays the definitions of the relevant symbols.

Based on (1) to (4) and by substituting symbols given in Table 2 into the supply chain process from C to E, the following formulas (5) to (10) are obtained:

**Table 3. Variable settings in Scenario 1.**

| | Symbol | Meaning | Symbol | Meaning |
|---|---|---|---|---|
| Cost related | $P_F$ | Unit freight rate | $P_D$ | Unit price of polymer desktop printing |
| | $P_I$ | Unit price of polymer industrial printing | | |
| | $C_D$ | Cost of desktop printer post-processing and local express | $C_I$ | Cost of industrial printer post-processing |
| Time related | $S_F$ | Freight speed | $S_D$ | Speed of polymer desktop printing |
| | $S_I$ | Speed of polymer industrial printing | | |
| | $D_{CE}$ | Distance from C to E | $D_{CF}$ | Distance from C to F |
| | $D_{FE}$ | Distance from F to E | | |
| | $T_D$ | Time of desktop printer post-processing and local express | $T_I$ | Time of industrial printer post-processing |
| Quality related variable | $E_D$ | Quality experience in desktop printing | $E_I$ | Quality experience in industrial printing |
| Quantity | $Q_P$ | Polymer printing material quantity | | |
| Weight | $\alpha$ | Weight of the cost utility | $\beta$ | Weight of the time utility |
| | $\gamma$ | Weight of the quality utility | | |
| Utility | $U_{D1}$ | Total utility of private-live-3DP in Scenario 1 | $U_{I1}$ | Total utility of corporate-live-3DP in Scenario 1 |

(5) Cost of private-live-3DP mode from C to E = $P_F * Q_P * D_{CE} + P_D * Q_P + C_D * Q_P$

(6) Time of private-live-3DP mode from C to E = $D_{CE} /S_F + Q_P / S_D + T_D * Q_P$

(7) Quality experience in private-live-3DP mode = $E_D$

(8) Cost of corporate-live-3DP mode from C to E = $P_F * Q_P * D_{CF} + P_I * Q_P + C_I * Q_P + P_F * Q_P * D_{FE}$

(9) Time of corporate-live-3DP mode from C to E = $D_{CF} / S_F + Q_P /S_I + T_I * Q_P + D_{FE} /S_F$

(10) Quality experience in corporate-live-3DP mode = $E_I$

The weight of the cost utility is denoted as α, the weight of time utility is denoted as β, and the weight of quality utility is denoted as γ. For consumers, the higher the printing cost, the lower the cost utility. Similarly, the longer the printing time, the lower the time utility. The better the quality, the higher the quality utility. Accordingly, $|\alpha|+|\beta|+|\gamma| = 1$.

Therefore, in the supply chain process from C to E, the total utilities of private-live-3DP and corporate-live-3DP modes are respectively given by:

$$U_{D1} = \alpha*(P_F*Q_P*D_{CE} + P_D*Q_P + C_D*Q_P) + \beta*(D_{CE}/S_F + Q_P/S_D + T_D*Q_P) + \gamma*E_D$$

$$U_{I1} = \alpha*(P_F*Q_P*D_{CF} + P_I*Q_P + C_I*Q_P + P_F*Q_P*D_{FE}) + \beta*(D_{CF}/S_F + Q_P/S_I + T_I*Q_P + D_{FE}/S_F) + \gamma*E_I$$

## 2.4 Scenario 2

In Scenario 2, both polymers and metals are used to produce finished products.

**2.4.1 Printing process.** Table 4 displays the settings of participants in the printing process.

Since we assume that a desktop printer cannot print metal parts, in Scenario 2, all metal parts are manufactured by industrial printers in F. The polymer can be processed using a desktop printer in E or an industrial printer in F. The specific settings are as follows.

Step 1: The starting points of Scenario 2 are A and B.

Step 2: Raw polymer is transported from A to C and processed into polymer printing filaments. Raw metal is transported from B to D and processed into metal printing powder. The powder is a common form of metal printing consumables.

Step 3: There are different material flow routes for different modes (private-live-3DP or corporate-live-3DP). Regardless of the route, consumer goods will eventually be generated at destination E.

If the polymer parts are produced by the private-live-3DP mode and the metal parts are produced by the corporate-live-3DP, the private live team purchases polymer printing filaments directly from C, and then prints and post-processes using the desktop printer in E to get

**Table 4. Participants in Scenario 2.**

| Symbol | Meaning | Symbol | Meaning |
|--------|---------|--------|---------|
| A | Raw polymer supplier | B | Raw metal supplier |
| C | Polymer printing filament producer | D | Metal printing powder producer |
| E | Private-live-3DP workshop/ End consumer | F | Corporate-live-3DP workshop |

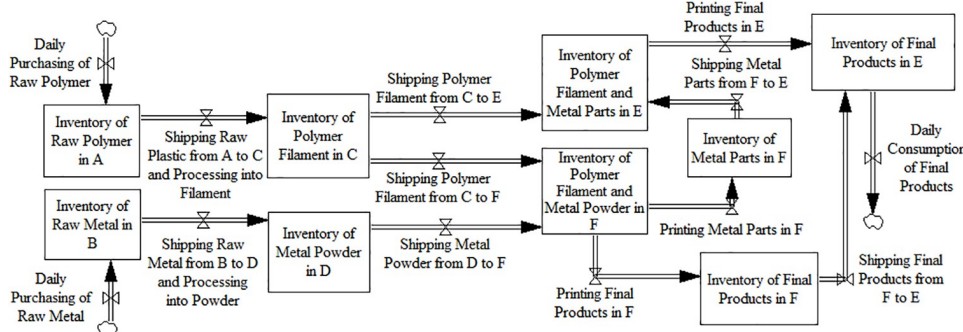

**Fig 2. System dynamics model of scenario 2.**

the polymer parts. Also, the corporate live team purchases metal printing powders from D, and then prints and post-processes with its own industrial printer to get the metal parts. Then, the metal parts are delivered to E, and the private live team assembles them together to produce the final products, then send them to the end consumer through local express.

If all parts are corporate-live-3DP, the corporate live team directly purchases polymer printing filaments from C and metal printing powders from D, and then prints and post-processes them into the final product using the industrial printer in F. Finally, the final product is delivered to E.

According to the flow routes, Fig 2 displays the system dynamics model of Scenario 2.

**2.4.2 Basic formulas.** According to Fig 2, the calculation formulas of cost, time, and quality of the routes are:

• Cost

(11) Cost of private-live-3DP mode = Freight of a raw polymer from A to C + Processing fee of the raw polymer in C + Freight of the polymer filaments from C to E + Printing fee of the polymer parts in E + Freight of raw metal from B to D + Processing fee of raw metal in D + Freight of the metal powders from D to F + Printing fee of the metal parts in F + Freight of the metal parts from F to E + Post-processing and local express fee of the final products in E.

(12) Cost of corporate-live-3DP mode = Freight of a raw polymer from A to C + Processing fee of the raw polymer in C + Freight of the polymer filaments from C to F + Printing fee of the polymer parts in F + Freight of raw metal from B to D + Processing fee of raw metal in D + Freight of the metal powders from D to F + Printing fee of the polymer parts in F + Post-processing fee of the final products in F + Freight of the final products from F to E.

• Time

(13) Time of private-live-3DP mode = Max (Transport time of a raw polymer from A to C + Processing time of the raw polymer in C + Transport time of the polymer filaments from C to E + Printing time of the polymer parts in E, Transport time of raw metal from B to D + Processing time of the raw metal in D + Transport time of the metal powders from D to F + Printing time of the metal parts in F + Transport time of the metal parts from F to E) + Post-processing and local express time of the final products in E.

(14) Time of corporate-live-3DP mode = Max (Transport time of a raw polymer from A to C + Processing time of the raw polymer in C + Transport time of the polymer filaments from

C to F + Printing time of the polymer parts in F, Transport time of the raw metal from B to D + Processing time of s raw polymer in D + Transport time of the metal powders from D to F + Printing time of the metal parts in F) + Post-processing time of the final products in F + Transport time of the final products from F to E.

- Quality

According to the above literature, the quality of the corporate-live-3DP product is at least as high as the quality of the private-live-3DP product.

**2.4.3 Total utility calculation.** Since the supply chain processes from A to C and from B to F are the same in Scenario 2, we need only compare the utilities of the private-live-3DP and corporate-live-3DP modes on the routes of polymer materials from C to E and metal materials from F to E.

Table 5 gives the definitions of the relevant symbols.

Based on (11) to (14) and by substituting symbols given in Table 4 into the supply chain process from C to E, the following formulas (15) to (20) are obtained:

(15) Cost of private-live-3DP mode from C to E = $P_F * Q_P * D_{CE} + P_D * Q_P + P_M * Q_M + P_F * Q_M * D_{FE} + C_D * (Q_P + Q_M)$

(16) Time of private-live-3DP mode from C to E = $Max (D_{CE} / S_F + Q_P / S_D, Q_M / S_M + D_{FE} / S_F) + T_D * (Q_P + Q_M)$

(17) Quality experience in private-live-3DP mode = $E_D$

(18) Cost of corporate-live-3DP mode from C to E = $P_F * Q_P * D_{CF} + P_I * Q_P + P_M * Q_M + P_F * (Q_P + Q_M) * D_{FE} + C_I * (Q_P + Q_M)$

(19) Time of corporate-live-3DP mode from C to E = $Max(D_{CF} / S_F + Q_P / S_I + D_{FE} / S_F, Q_M / S_M + D_{FE} / S_F) + T_I * (Q_P + Q_M)$

(20) Quality experience in corporate-live-3DP mode = $E_I$

Similar to Scenario 1, in Scenario 2, the weight of cost utility is α, the weight of time utility is β, and the weight of quality utility is γ. The correlation between the cost and cost utility is negative, as is the correlation between the time and time utility; whereas the correlation between the quality and quality utility is positive. Accordingly, $|α|+|β|+|γ| = 1$.

**Table 5. Variable settings in Scenario 2.**

| | Symbol | Meaning | Symbol | Meaning |
|---|---|---|---|---|
| Cost related | $P_F$ | Unit freight rate | $P_D$ | Unit price of polymer desktop printing |
| | $P_I$ | Unit price of polymer industrial printing | $P_M$ | Unit price of metal industrial printing |
| | $C_D$ | Cost of desktop printer post-processing and local express | $C_I$ | Cost of industrial printer post-processing and local express |
| Time related | $S_F$ | Freight speed | $S_D$ | Speed of polymer desktop printing |
| | $S_I$ | Speed of polymer industrial printing | $S_M$ | Speed of metal industrial printing |
| | $D_{CE}$ | Distance from C to E | $D_{CF}$ | Distance from C to F |
| | $D_{FE}$ | Distance from F to E | | |
| | $T_D$ | Time of desktop printer post-processing and local express | $T_I$ | Time of industrial printer post-processing and local express |
| Quality related variable | $E_D$ | Quality experience in desktop printing | $E_I$ | Quality experience in industrial printing |
| Quantity | $Q_P$ | Polymer printing material quantity | $Q_M$ | Metal printing material quantity |
| Weight | α | Weight of the cost utility | β | Weight of the time utility |
| | γ | Weight of the quality utility | | |
| Utility | $U_{D2}$ | Total utility of private-live-3DP printing in Scenario 2 | $U_{I2}$ | Total utility of corporate-live-3DP in Scenario 2 |

Therefore, in the supply chain process of a polymer from C to E and a metal from F to E, the total utilities of the private-live-3DP and corporate-live-3DP modes are respectively given by:

$$U_{D2} = \alpha*[P_F*Q_P*D_{CE} + P_D*Q_P + P_M*Q_M + P_F*Q_M*D_{FE} + C_D*(Q_P + Q_M)]$$
$$+ \beta*[Max(D_{CE}/S_F + Q_P/S_D, Q_M/S_M + D_{FE}/S_F) + T_D*(Q_P + Q_M)] + \gamma*E_D$$

$$U_{I2} = \alpha*[P_F*Q_P*D_{CF} + P_I*Q_P + P_M*Q_M + P_F*(Q_P + Q_M)*D_{FE} + C_I*(Q_P + Q_M)]$$
$$+ \beta*[Max(D_{CF}/S_F + Q_P/S_I + D_{FE}/S_F, Q_M/S_M + D_{FE}/S_F) + T_I*(Q_P + Q_M)] + \gamma*E_I$$

## 3. Optimize calculation

### 3.1 Optimal layout for Scenario 1

**3.1.1 Core variable of layout optimization.** Consumers have various preferences for different types of product utilities. Namely, some consumers pursue cheaper printed products, some pursue products with a faster access, and some require high-quality products. This affects the location of the utility equilibrium. According to the calculations of $U_{I1}$ and $U_{D1}$, the optimal utility brought to consumers by the two modes in Scenario 1 is related to the positions of C, E, and F. This means determining the optimal supply chain layout, with the core variables used to determine layout optimization being the location relationship among C, E, and F. Let this variable be Z, so that Z represents the relative positional variables of C, E, and F.

Set $Z = D_{CF}+D_{FE} - D_{CE}$. When the value of Z increases or decreases, the reasonable position of the printing platform changes, accordingly. Z is the core variable for determining whether to use the private live-3DP or corporate live-3DP mode. To explain the reason, construct a triangle having C, E, F as endpoints and side lengths $D_{CF}$, $D_{FE}$, and $D_{CE}$, respectively. When the value of Z increases, the distance between F and E becomes longer, and the construction of a central platform for the corporate-live-3DPbecomes more feasible. When the value of Z decreases or tends to zero, F will be on the $C_E$ segment, and it will be more feasible to build a local platform for the private-live-3DP. In Fig 3, the solid arrows indicate the direction of material flow.

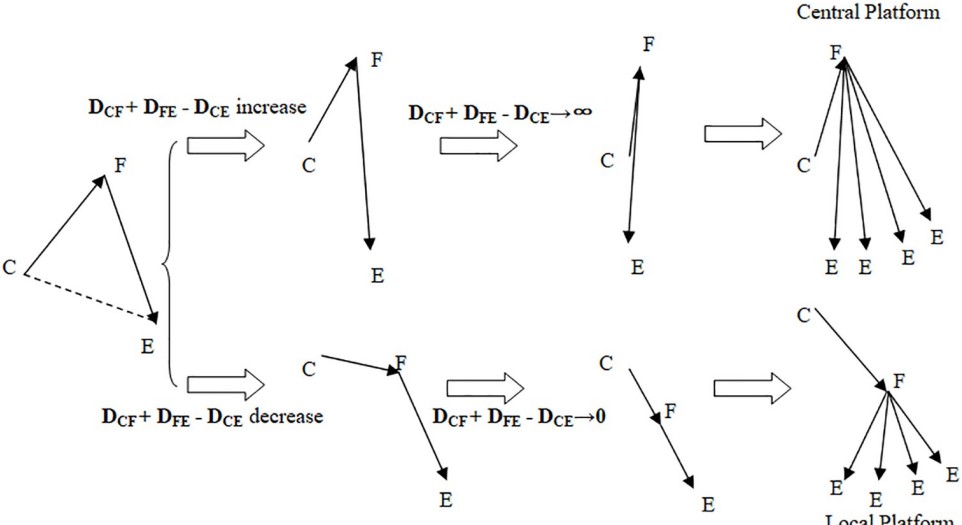

**Fig 3. Trends of $(D_{CF} + D_{FE} - D_{CE})$ when it tends to be maximum or minimum.**

When the utility of the two modes is equal, Z is in a state of utility balance. At this point, $U_{I1} = U_{D1}$, then:

$\alpha * (P_F * Q_P * D_{CE} + P_D * Q_P + C_D * Q_P) + \beta * (D_{CE} / S_F + Q_P / S_D + T_D * Q_P) + \gamma * E_D = \alpha * (P_F * Q_P * D_{CF} + P_I * Q_P + C_I * Q_P + P_F * Q_P * D_{FE}) + \beta * (D_{CF} / S_F + Q_P / S_I + T_I * Q_P + D_{FE} / S_F) + \gamma * E_I$

$\Leftrightarrow$

$U_{I1} - U_{D1} = (D_{CF} + D_{FE} - D_{CE}) * (\alpha * P_F * Q_P + \beta / S_F) + \alpha * Q_P * (P_I - P_D) + \alpha * Q_P * (C_I - C_D) + \beta * Q_P * (S_D - S_I) / (S_D * S_I) + \beta * Q_P * (T_I - T_D) + \gamma * (E_I - E_D) = 0$

$\Leftrightarrow$

(21) $Z = D_{CF} + D_{FE} - D_{CE} = -[\alpha * Q_P * (P_I - P_D + C_I - C_D) + \beta * Q_P * [(S_D - S_I) / (S_D * S_I) + T_I - T_D)] + \gamma * (E_I - E_D)] / (\alpha * P_F * Q_P + \beta / S_F)$

Formula (21) shows the location relationship among C, E, and F in Scenario 1 when the total utility of the corporate-live-3DP mode equals that of the private-live-3DP mode.

**3.1.2 Cost oriented layout optimization.** When consumers value the cost utility more, then:

$$|\alpha| \to 1, |\beta| + |\gamma| \to 0 \Leftrightarrow Z \to (P_D - P_I + C_D - C_I)/P_F.$$

When the weight of the cost utility tends to one, the positions of C, E, and F are mainly affected by corporate-live-3DP cost, private-live-3DP cost, and unit freight. The larger the value of $(P_D - P_I + C_D - C_I)$, which represents the cost difference between the corporate-live-3DP and private-live-3DP modes, the larger the value of Z. Similarly, the smaller the unit freight $P_F$, the larger the value of Z.

**3.1.3 Time oriented layout optimization.** When consumers value the time utility more, then:

$$|\beta| \to 1, |\alpha| + |\gamma| \to 0 \Leftrightarrow Z \to S_F * Q_P * [(S_I - S_D)/(S_D * S_I) + T_D - T_I)].$$

When the weight of the time utility tends to one, the positions of C, E, and F are mainly affected by the transport time, printing quantity, printing time, and post-processing time of the corporate-live-3DP and the private-live-3DP modes. If the value of $[(S_I - S_D) / (S_D * S_I) + T_D - T_I)]$ increases, which represents the difference in the printing and post-processing times between the corporate-live-3DP and private-live-3DP modes, the value of Z will also increase. Similarly, the greater the transport speed $S_F$, or the greater the printing quantity $Q_P$, the larger the value of Z.

**3.1.4 Quality oriented layout optimization.** When consumers value the quality utility more, then:

$$|\gamma| \to 1, |\alpha| + |\beta| \to 0 \Leftrightarrow \alpha * P_F * Q_P + \beta / S_F \to 0, Z \to \infty$$

When the weight of the quality utility tends to one, the value of Z tends towards infinity.

**3.1.5 Conclusions of optimal layout for Scenario 1.** Consequently, the following conclusions can be drawn:

- When the transportation speed is fast, or the printing quantity is preferred, or the unit freight is smaller, the value of Z will be large; therefore, the layout tends to the corporate-live-3DP mode and vice versa to the private-live-3DP mode.

- When the cost difference between the corporate-live-3DP and the private-live-3DP mode is large, the value of Z will be large; therefore, the layout tends to the corporate-live-3DP mode and vice versa to the private-live-3DP mode.

- When the difference in the printing and post-processing times between the corporate-live-3DP and private-live-3DP mode is large, the value of Z will be larger; therefore, the layout tends to the corporate-live-3DP mode and vice versa to the private-live-3DP mode.

- When consumers prefer high product quality, $Z \rightarrow \infty$, so the layout tends to the corporate-live-3DP mode and vice versa to the private-live-3DP mode.

## 3.2 Optimal layout for Scenario 2

### 3.2.1 Core variable of layout optimization.

According to the calculations of $U_{I2}$ and $U_{D2}$, the optimal utility brought to consumers by the two modes in Scenario 2 is also related to the positions of C, E, and F. This means that the core variable used to determine the optimization of supply chain layout is still Z, the location relationship among C, E, and F, and $Z = D_{CF} + D_{FE} - D_{CE}$. This is consistent with the setting of Scenario 1.

Consistent with Scenario 1's optimization approach, in Scenario 2, when $U_{I2} = U_{D2}$, then:

$$\alpha * [P_F * Q_P * D_{CE} + P_D * Q_P + P_M * Q_M + P_F * Q_M * D_{FE} + C_D * (Q_P + Q_M)] + \beta * [Max(D_{CE} / S_F + Q_P / S_D, Q_M / S_M + D_{FE} / S_F) + T_D * (Q_P + Q_M)] + \gamma * E_D = \alpha * [P_F * Q_P * D_{CF} + P_I * Q_P + P_M * Q_M + P_F *(Q_P + Q_M) * D_{FE} + C_I * (Q_P + Q_M)] + \beta * [Max(D_{CF}/S_F + Q_P / S_I + D_{FE} / S_F, Q_M / S_M + D_{FE} / S_F) + T_I * (Q_P + Q_M)] + \gamma * E_I$$

$\Leftrightarrow$

$$U_{I2} - U_{D2} = \alpha * P_I * Q_P + \alpha * P_M * Q_M + \alpha * P_F * D_{CF} * Q_P + \alpha * P_F * D_{FE} * Q_P + \alpha * P_F * D_{FE} * Q_M + \alpha * C_I * Q_P + \alpha * C_I * Q_M + \beta * Max(D_{CF} / S_F + Q_P / S_I + D_{FE} / S_F, Q_M / S_M + D_{FE} / S_F) + \beta * T_I * Q_P + \beta * T_I * Q_M + \gamma * E_I - \alpha * P_F * Q_P * D_{CE} - \alpha * P_D * Q_P - \alpha * P_M * Q_M - \alpha * P_F * Q_M * D_{FE} - \alpha * C_D * Q_P - \alpha * C_D * Q_M - \beta * Max(D_{CE} / S_F + Q_P / S_D, Q_M / S_M + D_{FE} / S_F) - \beta * T_D * Q_P - \beta * T_D * Q_M - \gamma * E_D$$

$\Leftrightarrow$

$$U_{I2} - U_{D2} = \alpha * Q_P * (P_I - P_D) + \alpha * P_F * Q_P * Z + \alpha * (Q_P + Q_M) * (C_I - C_D) + \beta * Max [Z / S_F + Q_P * (1 / S_I - 1 / S_D), 0] + \beta * (Q_P + Q_M) * (T_I - T_D) + \gamma * (E_I - E_D)$$

According to the current technical performances of desktop and industrial printers, in general, it holds that: $P_I > P_D$, $C_I > C_D$, $S_I > S_D$, $T_I < T_D$, $E_I > E_D$, $\Leftrightarrow (1 / SI - 1/ S_D) < 0$.

Because $Z \geq 0$,

$\Leftrightarrow Z / S_F + Q_P * (1 / S_I - 1 / S_D)$ can be greater than zero or less than or equal to zero.

When $Z / S_F + Q_P * (1 / S_I - 1 / S_D) > 0$,

$$\alpha * Q_P * (P_I - P_D) + \alpha * P_F * Q_P * Z + \alpha * (Q_P + Q_M) * (C_I - C_D) + \beta * [Z / S_F + Q_P * (1 / S_I - 1 / S_D)] + \beta * (Q_P + Q_M) * (T_I - T_D) + \gamma * (E_I - E_D) = 0$$

$\Leftrightarrow$

$$(22) \quad Z = -[\alpha * Q_P * (P_I - P_D) + \alpha * (Q_P + Q_M) * (C_I - C_D) + \beta * Q_P *(1 / S_I - 1 / S_D) + \beta * (Q_P + Q_M) * (T_I - T_D) + \gamma * (E_I - E_D)]/[\alpha * P_F * Q_P + \beta / S_F]$$

When $Z / S_F + Q_P * (1 / S_I - 1 / S_D) \leq 0$,

$\alpha * Q_P * (P_I - P_D) + \alpha * P_F * Q_P * Z + \alpha * (Q_P + Q_M) * (C_I - C_D) + \beta * (Q_P + Q_M) * (T_I - T_D) + \gamma * (E_I - E_D) = 0$

$\Leftrightarrow$

(23) $Z = -[\alpha * Q_P * (P_I - P_D) + \alpha * (Q_P + Q_M) * (C_I - C_D) + \beta * (Q_P + Q_M) * (T_I - T_D) + \gamma * (E_I - E_D)] / \alpha * P_F * Q_P$

Formula (23) shows the location relationship among C, E, and F in Scenario 2 when the total utility of the corporate-live-3DP mode equals that of the private-live-3DP mode. For different types of utilities, the influence of the weight change on the location of the utility equilibrium point is as follows.

**3.2.2 Cost oriented layout optimization.** When consumers value the cost utility more, then:

$$|\alpha| \to 1, |\beta| + |\gamma| \to 0 \Leftrightarrow Z \to (P_D - P_I)/P_F + (Q_P + Q_M)*(C_D - C_I)/P_F*Q_P$$

When the weight of the cost utility tends to one, whether $[Z/S_F + Q_P * (1/S_I - 1/S_D)]$ is greater than zero, the locations of C, E, and F are mainly affected by the corporate-live-3DP cost, private-live-3DP cost, and the unit freight. As the value of $(P_D - P_I + C_D - C_I)$ increases, Z will also increase. Similarly, the smaller the unit freight $P_F$, the larger the value of Z. This is the same as for the printing process with just polymer.

In addition, the quantity ratio of metal printing to polymer printing also affects the position of C, E, and F. The larger the ratio $Q_M/Q_P$, the greater the value of Z. This demonstrates that the best layout will be affected by the interaction when the process contains two different printing materials, which differs from the process when printing is conducted using only the polymer.

**3.2.3 Time oriented layout optimization.** When consumers value the time utility more, then:

$|\beta| \to 1, |\alpha| + |\gamma| \to 0 \Leftrightarrow Z \to S_F * Q_P * [(S_I - S_D) / (S_D * S_I) + T_D - T_I]$.

If $Z / S_F + Q_P * (1 / S_I - 1 / S_D) > 0 \Leftrightarrow Z \to S_F * [Q_P * (1 / S_D - 1 / S_I) + (Q_P + Q_M) * (T_D - T_I)]$

If $Z / S_F + Q_P * (1 / S_I - 1 / S_D) < 0 \Leftrightarrow Z \to \infty$

When $[Z / S_F + Q_P * (1 / S_I - 1 / S_D)] < 0$, and the weight of the time utility tends to one, the positions of C, E, and F are mainly affected by the transport time, printing quantity, printing time, and the post-processing time of the corporate-live-3DP and private-live-3DP modes. The larger the value of $[(S_I - S_D) / (S_D * S_I) + T_D - T_I]$, the larger the value of Z. Similarly, the greater the transport speed SF, or the greater the printing quantity QP, the larger the value of Z.

In addition, the quantity ratio of metal printing to polymer printing also affects the position of C, E, and F. The larger QM/QP, the greater the value of Z. This comports with the characteristic: when $|\alpha| \to 1$.

When $[Z / SF + QP * (1 / S_I - 1 / S_D)] < 0$, Z tends towards infinity.

**3.2.4 Quality oriented layout optimization.** When consumers value the quality utility more, then:

• $|\gamma| \to 1, |\alpha| + |\beta| \to 0 \Leftrightarrow \alpha * P_F * Q_P + \beta / S_F \to 0, Z \to \infty$

When the weight of the quality utility tends to one, then the value of Z tends towards infinity.

**3.2.5 Conclusions of optimal layout for Scenario 2.** Consequently, the following conclusions can be drawn:

- When the transportation speed is fast, or the printing quantity is preferred, or the unit freight is small, the value of Z will be large; therefore, the layout tends to the corporate-live-3DP mode and vice versa to the private-live-3DP mode.

- When the cost difference between the corporate-live-3DP and private-live-3DP modes is large, the value of Z will be large; therefore, the layout tends to the corporate-live-3DP mode and vice versa to the private-live-3DP mode.

- When the difference in printing and post-processing times of the corporate-live-3DP and private-live-3DP modes is large, the value of Z will also be large; thus, the layout tends to the corporate-live-3DP mode and vice versa to the private-live-3DP mode.

- When consumers prefer product quality, $Z \to \infty$, and the layout tends to the the corporate-live-3DP mode and vice versa to the private-live-3DP mode.

- In the 3DP hybrid printing process, it is assumed that the metal can only be printed by an industrial printer. If the ratio of metal to polymer printing number is large, the value of Z may also be large, and the layout tends to the corporate-live-3DP mode and vice versa to the private-live-3DP mode.

# 4. Discussion

According to the current trend of live streaming e-commerce, based on different printing materials and modes, this research deduced an optimal production layout. The main findings are concluded in the Table 6.

According to Table 6, in the current stage of technological development, if only considering cost, time, and quality utility, whether it is pure polymer printing or mixed printing of polymer and metal, the performance of the Corporate-live-3DP mode is better. This means that the entire industry will develop towards a direction where Corporate leads the entire live-3DP business. However, according to existing literature [40–42], Private-live-3DP still has a huge customer base without pursuing optimal utility. Under the influence of two modes, the entire live-3DP supply chain may undergo the following changes.

## 4.1 Supply chain optimization for the transition of live streaming e-commerce to the corporate-live-3DP model

**4.1.1 Consumer analysis of the corporate live-3DP model.** Live e-commerce programs adopt enterprise 3D printing mode to provide product customization services, mainly targeting the following target consumers:

- Pursuing professional quality

Table 6. Comparison of supply chain layout optimization between Scenario 1 and Scenario 2.

|  | Scenario 1 | Scenario 2 |
|---|---|---|
| Applicable materials | Polymer | Polymer and Metal |
| Conducive to reducing manufacturing costs | Corporate-live-3DP | Corporate-live-3DP |
| Conducive to reducing manufacturing time | Corporate-live-3DP | Corporate-live-3DP |
| Conducive to improving quality | Corporate-live-3DP | Corporate-live-3DP |

Enterprise level 3D printing equipment ensures product professionalism and consistency, official certification and brand endorsement enhance purchasing confidence, especially suitable for consumers who trust well-known brands and seek quality assurance.

- Large scale demand and rapid response

Large scale production and rapid response capabilities are suitable for large audiences, high demand scenarios, and consumers who require high timeliness and participate in large-scale promotional activities.

- Pay attention to regulatory compliance and standard compliance

A comprehensive quality management system should address complex regulatory and certification requirements, ensure product compliance, reduce legal risks, and attract consumers who are sensitive to compliance.

- Emphasize supply chain integration and efficiency

Relying on economies of scale and supply chain synergy advantages, integrate resources, reduce costs, shorten delivery cycles, provide cost-effective goods, and attract consumers who pursue shopping efficiency and value supply chain performance.

- Prefer high-end, professional, and authoritative shopping experiences

High quality, standardized products and services support high-end, professional, or authoritative program positioning, cooperate with well-known 3D printing enterprises to enhance industry influence and audience professional awareness, and attract consumers who pursue high-end experiences, value brand reputation, and professional services.

**4.1.2 Optimization of the corporate live-3DP model on the supply chain.** The corporate-live-3DP model has a significant impact on traditional supply chains, reflected in the improvement of supply chain agility and flexibility, global layout and efficient distribution, enhanced innovation driving force and rapid product iteration, inventory cost reduction and refined management, capacity planning and coordination challenges, balance between quality control and standardized processes, supply chain transparency and consumer right to know. In order to adapt to the impact of the Corporate live-3DP model, traditional supply chains can be optimized from the following aspects:

- Technological upgrading and digital transformation

Introducing advanced supply chain management systems (SCM) to achieve real-time tracking and intelligent analysis of orders, inventory, logistics and other information, and improve decision-making efficiency. Utilize technologies such as the Internet of Things (IoT), big data, and artificial intelligence to enhance the perception ability of the supply chain, accurately predict market demand, and quickly respond and adjust production plans. Establish a cloud platform or blockchain system to achieve full traceability of the supply chain, enhance transparency, and meet the right to information needs of consumers regarding product sources, production processes, material composition, and other information.

- Flexibility and Agile Production

Gradually introducing modular and reconfigurable production line designs to quickly adapt to product design changes and fluctuations in order demands. Promote the concept of lean production, eliminate waste, shorten production cycles, and improve the response speed of production systems. Collaborate with 3D printing service providers to outsource the

production tasks of some customized, small batch or complex components to 3D printing centers, leveraging their fast response and on-demand production capabilities.

- Global Supply Chain Network Optimization

Re examine the global supply chain layout, and reasonably set or adjust production bases, warehouses, and distribution centers based on market demand, costs, risks, and other factors to achieve nearby production and distribution, reduce logistics costs and delivery times. Strengthen collaboration with global 3D printing centers, utilize their distributed manufacturing capabilities as a supplement or buffer to traditional supply chains, and respond to sudden and regional order demands [43].

- Innovation management and collaborative research and development:

Establish a close interaction mechanism with consumers, designers, and R&D teams, quickly collect and respond to consumer feedback, and promote rapid product iteration. Collaborate with 3D printing technology research and development institutions, universities, etc. to jointly explore new materials and processes, and promote supply chain technology innovation. Strengthen collaborative research and development with suppliers and partners, jointly respond to market changes, share innovative achievements, and enhance the innovation driving force of the overall supply chain.

- Refined inventory management:

Implement precise prediction and intelligent replenishment, reduce inventory backlog and overproduction, and optimize inventory structure and level. By adopting advanced inventory management models such as JIT (Just In Time) or VMI (Vendor Managed Inventory), real-time demand information is shared with suppliers to achieve dynamic adjustment and refined management of inventory.

- Upgrade of quality control system:

Strengthen quality monitoring throughout the entire process to ensure that temporary adjustments do not affect product quality, especially in new materials and process links related to 3D printing. Collaborate with 3D printing service providers to establish and implement strict quality standards and certification processes, ensuring that the quality of outsourced products is consistent with that of in-house products. Regularly conduct supply chain risk assessments and compliance audits to promptly identify and resolve potential quality and compliance issues.

- Talent cultivation and organizational structure adjustment:

Cultivate professional talents with digital thinking, familiar with 3D printing technology and supply chain management, and enhance the team's ability to respond to new technological challenges. Adjust the organizational structure and establish a dedicated supply chain innovation department or project team responsible for promoting the application of 3D printing technology and supply chain optimization work. Establish cross departmental and cross organizational collaboration mechanisms, break down departmental barriers, and improve the overall collaborative efficiency of the supply chain.

Through the above optimization measures, traditional supply chains can better adapt to the impact of enterprise 3D printing mode, improve the agility, flexibility, efficiency, and transparency of the supply chain, meet the needs of consumers for personalization, rapid response, high quality, environmental protection, and other aspects, and enhance the competitiveness of enterprises.

## 4.2 Supply chain optimization for the transition of live streaming e-commerce to the private-live-3DP model

**4.2.1 Consumer analysis of the private-live-3DP model.** Live e-commerce programs adopt the Private-live-3DP model to provide product customization services, mainly attracting the following consumers:

• Pursuing personalization and uniqueness

Flexibly designing services to meet personalized needs, enhancing interactivity and uniqueness through live streaming design and printing processes, and providing exclusive customized products.

• Emphasize community participation and emotional connection

Support independent designers, craftsmen, or niche brands, value direct communication with creators, establish emotional connections, and turn the shopping process into emotional communication and cultural experience.

• Favor innovative experimentation and market testing

Fast trial and error, innovative design and market validation, adapting to emerging, rapid iteration or creative oriented industries, and meeting the demand for novel, unique, and cutting-edge products.

• Focus on environmental protection and sustainability

Small batch, on-demand production reduces resource waste, is easy to use environmentally friendly materials and processes, and meets the expectations of green and sustainable consumption.

• Pursuing content innovation and deep interaction

As a distinctive content element, designers showcase the design and printing process on site, increasing visual and educational value, attracting audiences interested in 3D printing technology, and enhancing participation and stickiness.

• Serving specific interest groups and niche markets

Providing diversified products that meet niche needs, filling market gaps, enhancing the reputation and influence of specific audience groups, and accurately matching personalized needs of segmented markets.

**4.2.2 Optimization of the private live-3DP model on the supply chain.** The Private-live-3DP model has multiple impacts on traditional supply chains, including improved supply chain response speed, distributed manufacturing and localized production, enhanced innovation driving force and rapid product iteration, reduced inventory costs and refined management, capacity bottlenecks and supply-demand balance challenges, quality control and standardized processes, supply chain transparency and consumer right to know. Some are similar to the Coporate-live-3DP model, while others are different. In order to adapt to the impact of the Private-live-3DP model, traditional supply chains can be optimized from the following aspects:

• Integrate private 3D printing resources

Establish a collaborative network with private printers and incorporate them into the supply chain system as a supplementary force for quickly responding to personalized orders, small-scale production, and localized services. Design a reasonable profit distribution

mechanism to encourage private printers to actively participate, while ensuring their service quality and compliance.

- Demand forecasting and order allocation optimization

Improve the accuracy of demand forecasting, combine data analysis tools to predict the trend of personalized and small batch orders, and provide a basis for capacity planning. Establish a flexible order allocation system that automatically matches suitable production resources (including traditional production lines, enterprise level 3D printing centers, and private printers) based on order characteristics (such as quantity, region, delivery time, etc.) to ensure supply-demand balance.

- Supply chain collaboration and information sharing platform

Build a digital supply chain collaboration platform to achieve real-time information exchange with private printers, including order transmission, design file transmission, production progress tracking, quality monitoring, etc. Through the platform, resource sharing and complementary capabilities can be achieved, such as design templates, material databases, process guides, etc., to help private printers improve production efficiency and quality.

- Extension of quality management system

Extend quality control standards and processes to private printers, ensuring compliance with unified quality requirements through training, certification, and other means. Implement remote quality monitoring and regular spot checks, and continuously improve quality based on user feedback. Establish a rapid response mechanism to promptly intervene and properly handle quality issues.

- Inventory strategy adjustment

Based on the characteristics of private 3D printing mode, adjust inventory strategy, reduce inventory of conventional products, increase inventory of general materials, semi-finished products, etc., to quickly respond to personalized orders. Implement dynamic inventory management, adjust inventory levels in real-time based on real-time order demand and the production capacity of private printers.

- Improvement of supply chain transparency

Share the requirements and standards for supply chain transparency construction with private printers, encourage them to use traceable materials, disclose production process information, etc. Integrate product information provided by private printers through digital platforms to ensure that consumers can access complete information on product sources, production processes, material composition, and more.

- Regulatory compliance and risk management

Provide necessary regulatory training and guidance to ensure that private printers understand and comply with relevant industry regulations and laws. Regularly assess the risk status of private printers, including intellectual property protection, data security, environmental compliance, etc., and take preventive measures to reduce risks.

Through the above optimization measures, traditional supply chains can effectively integrate private 3D printing resources, improve response speed to personalized and small batch orders, achieve distributed manufacturing and localization services, while ensuring product quality, supply chain transparency, and regulatory compliance, thereby adapting to the impact

of private 3D printing mode, improving the overall competitiveness and customer satisfaction of the supply chain.

# 5. Conclusion and future work

## 5.1 Conclusion

This research analyzes the impact of the increasingly popular live streaming e-commerce on the 3DP supply chain, and uses system dynamics methods to construct two supply chain models for pure polymer printing and polymer metal mixed printing, exploring the optimization layout of the 3DP supply chain. The conclusion shows that if only considering the equivalent indicators of printing time, printing cost, and printing quality, regardless of the materials used, the Corporate-live-3DP service is the best choice for live streaming e-commerce. However, in practical applications, combined with data and cases from other literature, Private-live-3DP still has a broad consumer base.

The main differences between Corporate live-3DP and Private live-3DP lie in performance, efficiency, cost control, and quality standards. Corporate live-3DP focuses on large-scale production, efficient operation, strict quality control, and long-term cost optimization, making it suitable for industrial manufacturing and mass customization. The Private-live-3DP model is more suitable for small-scale, personalized, low-cost, and highly flexible application scenarios, such as education, creative design, and home DIY. In actual selection, it is necessary to determine which mode is more suitable based on specific needs, budget, expected output, and quality standards.

Both models will have a significant impact on traditional supply chains. In order to adapt to this change, the former can be optimized through technological upgrades and digital transformation, flexible and agile production, global supply chain network optimization, innovative management and collaborative research and development, refined inventory management, quality control system upgrading, talent cultivation and organizational structure adjustment, and other means. The latter can be optimized through integration of private 3D printing resources, demand forecasting and order allocation optimization, supply chain collaboration and information sharing platform, extension of quality management system, inventory strategy adjustment, improvement of supply chain transparency, regulatory compliance and risk management, and other means.

## 5.2 Limitations and suggestion for future research

**5.2.1 Limitations.** There are still the following shortcomings and limitations in this study.

- Single utility indicator

This study only considered equivalent indicators of printing time, printing cost, and printing quality, and did not fully cover other important factors that affect the choice of 3DP mode in live streaming e-commerce, such as consumer preferences, brand positioning, market trends, environmental impact, etc.

- Model simplification

The constructed supply chain model focuses on two scenarios: pure polymer printing and polymer metal hybrid printing, and fails to cover all types of 3DP technologies and their applications in different industries and scenarios, which may limit the universality of the conclusions.

- Data dependence

The research conclusion partially relies on data and cases from other literature, and the timeliness, representativeness, and differences in research methods of these data may affect the accuracy and reliability of the conclusion.

- Not considering dynamic changes

The live streaming e-commerce market, 3DP technology, and consumer demand are rapidly changing, and this study has not fully explored the evolutionary trends of these dynamic factors on the future supply chain.

**5.2.2 Future works.** In subsequent research, the following directions can be approached to achieve better research results.

- Multi utility evaluation

Further research and incorporate more utility indicators that affect the choice of 3DP model in live streaming e-commerce, such as consumer satisfaction, brand value, environmental benefits, etc., to build a more comprehensive and three-dimensional evaluation system.

- Technology types and industry application expansion

Deepen research on various 3DP technologies (such as bioprinting, ceramic printing, etc.) and their applications in different industries (such as healthcare, construction, food, etc.), and build a broader supply chain model to adapt to diversified market demands.

- Real time data and empirical analysis

Utilize real-time market data and field research to conduct more in-depth empirical analysis, in order to enhance the timeliness and authenticity of research conclusions.

- Dynamic adaptability research

Exploring how the 3DP supply chain can adapt to the rapid changes in the live streaming e-commerce market, including the introduction of emerging technologies, the evolution of consumer behavior, and adjustments to regulatory policies, providing forward-looking guidance for the continuous optimization of the supply chain.

- Exploration of hybrid mode

Study the integration and complementary strategies of Corporate live-3DP and Private live-3DP modes in live streaming e-commerce, and explore how to build a hybrid supply chain that combines large-scale production efficiency and personalized customization flexibility to maximize the satisfaction of various consumer needs.

- Environmental and social impact assessment

Consider the impact of the 3DP supply chain on the environment (such as energy consumption and waste management) and society (such as employment and fair trade), and promote the formation of a more sustainable and responsible 3DP supply chain model for live streaming e-commerce.

## Acknowledgments

The authors thank LetPub (www.letpub.com) for its linguistic assistance and scientific consultation during the preparation of this manuscript.

## Author Contributions

**Conceptualization:** Zhen Chen.

**Data curation:** Zhen Chen, Ying Tang.

**Formal analysis:** Zhen Chen.

**Funding acquisition:** Zhen Chen.

**Investigation:** Ying Tang.

**Methodology:** Zhen Chen.

**Project administration:** Zhen Chen.

**Resources:** Zhen Chen.

**Software:** Zhen Chen, Ying Tang.

**Validation:** Ying Tang.

**Visualization:** Zhen Chen, Ying Tang.

**Writing – original draft:** Zhen Chen.

**Writing – review & editing:** Ying Tang.

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
