## [Decision Letter · Decision Letter 0]

26 Feb 2024

PONE-D-23-43486Optimization of 3D printing supply chain in the era of live streaming e-commercePLOS ONE

Dear Dr. Tang,

Thank you for submitting your manuscript to PLOS ONE. After careful consideration, we feel that it has merit but does not fully meet PLOS ONE’s publication criteria as it currently stands. Therefore, we invite you to submit a revised version of the manuscript that addresses the points raised during the review process. Please submit your revised manuscript by Apr 11 2024 11:59PM. If you will need more time than this to complete your revisions, please reply to this message or contact the journal office at plosone@plos.org. Please include the following items when submitting your revised manuscript:A rebuttal letter that responds to each point raised by the academic editor and reviewer(s). You should upload this letter as a separate file labeled 'Response to Reviewers'.A marked-up copy of your manuscript that highlights changes made to the original version. You should upload this as a separate file labeled 'Revised Manuscript with Track Changes'.An unmarked version of your revised paper without tracked changes. You should upload this as a separate file labeled 'Manuscript'.

We look forward to receiving your revised manuscript.

Kind regards,

Rehana Naz

Academic Editor

PLOS ONE

Journal Requirements:

2. Thank you for submitting the above manuscript to PLOS ONE. During our internal evaluation of the manuscript, we found significant text overlap between your submission and previous work in the [introduction, conclusion, etc.].

Please revise the manuscript to rephrase the duplicated text, cite your sources, and provide details as to how the current manuscript advances on previous work. Please note that further consideration is dependent on the submission of a manuscript that addresses these concerns about the overlap in text with published work.

[If the overlap is with the authors’ own works: Moreover, upon submission, authors must confirm that the manuscript, or any related manuscript, is not currently under consideration or accepted elsewhere. If related work has been submitted to PLOS ONE or elsewhere, authors must include a copy with the submitted article. Reviewers will be asked to comment on the overlap between related submissions (http://journals.plos.org/plosone/s/submission-guidelines#loc-related-manuscripts).]

We will carefully review your manuscript upon resubmission and further consideration of the manuscript is dependent on the text overlap being addressed in full. Please ensure that your revision is thorough as failure to address the concerns to our satisfaction may result in your submission not being considered further.

"1. Guangdong Philosophy and Social Science Planning Project, grant number GD22YDXZYJ02.

2. Special projects in key fields of Guangdong Provincial Department of Education, grant number 2021ZDZX3003."

Reviewers' comments:

Reviewer's Responses to Questions

**Comments to the Author**

1. Is the manuscript technically sound, and do the data support the conclusions?

Reviewer #1: No

Reviewer #2: Partly

2. Has the statistical analysis been performed appropriately and rigorously? 

Reviewer #1: N/A

Reviewer #2: N/A

3. Have the authors made all data underlying the findings in their manuscript fully available?

Reviewer #1: Yes

Reviewer #2: No

4. Is the manuscript presented in an intelligible fashion and written in standard English?

Reviewer #1: No

Reviewer #2: Yes

5. Review Comments to the Author

Reviewer #1: First of all, I appreciate the opportunity to review the paper Optimization of 3D printing supply chain in the era of live streaming e-commerce. The paper deals with a very interesting problem.

Suggestions are below:

The Abstract should be improved. The more about motivation, methods, results, and findings is missing. “Live streaming e-commerce” must be mentioned in the abstract.

The last paragraph in the introduction section should be a structure of the paper (several sentences for each section). This is missing.

It is necessary to understand the purpose and aim of the paper as well as its "position" in previous research (also gap analysis).

The separate section Practical and Theoretical Implications (or Discussion) is missing. The existing section Discussion is very modest. This confirms the lack of scientific and practical contribution.

The paper is not written in a scientific manner. For example, see Section 3. Explanations are missing.

The methodology is not clear and sufficiently explained. Argumentation is missing.

Results and findings are questionable.

Scientific contributions are also questionable.

Also, the conclusion section is not on a satisfactory level. The conclusion in scientific papers is very important.

o Limitations of your research must be emphasized

o Future research directions must be reinforced.

Suggested references

https://doi.org/10.1080/13675567.2022.2037125

https://doi.org/10.1108/EOR-01-2023-0004

Jiang, F., Tian, S. Q., Sremac, S., & Huskanović, E. (2023). Analyzing Traceability Models in E-Commerce Logistics: A Multi- Channel Approach. J. Ind Intell., 1(4), 203-218. https://doi.org/10.56578/jii010402

Fazlollahtabar, H. (2022). Mathematical Modeling for Sustainability Evaluation in a Multi-Layer Supply Chain. J. Eng. Manag. Syst. Eng., 1(1), 2-14. https://doi.org/10.56578/jemse010102

Reviewer #2: In this paper, the authors have explored the uniqueness of the 3D printing supply chain for metal and non-metal products, and calculated the utility equilibrium point between private and corporate live streaming 3D printing. The research results show that corporate live 3D printing mode is more suitable for both polymer printing and polymer-metal hybrid printing. But private live 3D printing mode can still meet the needs of some consumers. This change will not only create a new consumption model, but also reshape our global economic system, making it more resilient, flexible and localized. This will also stimulate the further development of 3D printing technology.

In my opinion, the idea of the manuscript is interesting and can be considered as a candidate for publication in the journal. However, to further justify why this paper should be published, the authors must address the following queries and comments:

1. The current abstract should be greatly improved as it does not reveal the novelty of the current study.

2. Can the authors provide more details on how the utility equilibrium point between private and corporate live streaming 3D printing was calculated?

3. What specific criteria were considered when comparing the suitability of corporate and private live 3D printing modes for polymer and polymer-metal hybrid printing?

4. Could you elaborate on the analysis of consumer needs that led to the conclusion that private live 3D printing mode can still meet the needs of some consumers?

5. The introduction of Section 3.1 is very poor. Improve it.

6. In what ways does the transition to a corporate live 3D printing mode create a new consumption model, and how is this different from the current models?

7. How does the proposed shift in 3D printing modes contribute to making the global economic system more resilient, flexible, and localized? Can you provide specific examples or mechanisms?

8. What are the potential implications for localized production and supply chain dynamics as a result of the shift to a corporate live 3D printing mode?

9. In terms of comparison with previous studies, the discussion is not acceptable. It must be greatly improved during revision.

10. Can you elaborate on how the proposed changes are expected to stimulate further development in 3D printing technology? Are there specific areas or aspects that will see accelerated growth?

11. Did the study take into account the environmental impact of the different 3D printing modes, especially considering the increasing focus on sustainability in manufacturing?

12. What limitations or constraints were encountered during the research, and how might they have influenced the study's findings and conclusions?

13. How do the research results suggest potential changes in industry practices related to 3D printing, and are there specific recommendations for practitioners based on the findings?

6. PLOS authors have the option to publish the peer review history of their article (what does this mean?). If published, this will include your full peer review and any attached files.

Reviewer #1: No

Reviewer #2: No

---

## [Author Response · Author response to Decision Letter 0]

10 Apr 2024

We have provided a detailed response to your guidance in the document 《Respond to Reviews》. We spent over a month revising 70% of the text in the manuscript. We hope that could meet your requirements. Thank you.

---

## [Decision Letter · Decision Letter 1]

22 Apr 2024

Optimization of 3D printing supply chain in the era of live streaming e-commerce

PONE-D-23-43486R1

Dear Dr. Tang,

We’re pleased to inform you that your manuscript has been judged scientifically suitable for publication and will be formally accepted for publication once it meets all outstanding technical requirements.

Kind regards,

Rehana Naz

Academic Editor

PLOS ONE

Additional Editor Comments (optional):

Reviewers' comments:

Reviewer's Responses to Questions

**Comments to the Author**

1. If the authors have adequately addressed your comments raised in a previous round of review and you feel that this manuscript is now acceptable for publication, you may indicate that here to bypass the “Comments to the Author” section, enter your conflict of interest statement in the “Confidential to Editor” section, and submit your "Accept" recommendation.

Reviewer #1: All comments have been addressed

Reviewer #2: All comments have been addressed

2. Is the manuscript technically sound, and do the data support the conclusions?

Reviewer #1: Yes

Reviewer #2: Yes

3. Has the statistical analysis been performed appropriately and rigorously? 

Reviewer #1: Yes

Reviewer #2: N/A

4. Have the authors made all data underlying the findings in their manuscript fully available?

Reviewer #1: Yes

Reviewer #2: Yes

5. Is the manuscript presented in an intelligible fashion and written in standard English?

Reviewer #1: Yes

Reviewer #2: Yes

6. Review Comments to the Author

Reviewer #1: The paper should be accepted. All comments from the first round are done. No additional suggestion. Congratulations!

Reviewer #2: All my queries have been addressed by the authors. The revised version can be accepted for publication.

7. PLOS authors have the option to publish the peer review history of their article (what does this mean?). If published, this will include your full peer review and any attached files.

Reviewer #1: No

Reviewer #2: No

---

## [Editor Report · Acceptance letter]

3 May 2024

PONE-D-23-43486R1 

PLOS ONE

Dear Dr. Tang, 

I'm pleased to inform you that your manuscript has been deemed suitable for publication in PLOS ONE. Congratulations! Your manuscript is now being handed over to our production team.

Kind regards, 

on behalf of

Prof Rehana Naz 

Academic Editor

PLOS ONE